# Temporal and Spatial Variation of Aboveground Biomass of *Pinus densata* and Its Drivers in Shangri-La, CHINA

**DOI:** 10.3390/ijerph19010400

**Published:** 2021-12-30

**Authors:** Dongfan Xu, Jialong Zhang, Rui Bao, Yi Liao, Dongyang Han, Qianwei Liu, Tao Cheng

**Affiliations:** 1Faculty of Forestry, Southwest Forestry University, Kunming 650224, China; xdf100kg@swfu.edu.cn (D.X.); baorui@swfu.edu.cn (R.B.); ianliao@swfu.edu.cn (Y.L.); handy@swfu.edu.cn (D.H.); 2Faculty of Geography, Yunnan Normal University, Kunming 650092, China; liuqw@user.ynnu.edu.cn; 3National Geomatics Center of China, Beijing 100089, China; chengtao@ngcc.cn

**Keywords:** *Pinus densata*, aboveground biomass changes, structural equation model, driving force

## Abstract

Understanding the drivers of forest aboveground biomass (AGB) is essential to further understanding the forest carbon cycle. In the upper Yangtze River region, where ecosystems are incredibly fragile, the driving factors that make AGB changes differ from other regions. This study aims to investigate AGB’s spatial and temporal variation of *Pinus densata* in Shangri-La and decompose the direct and indirect effects of spatial attribute, climate, stand structure, and agricultural activity on AGB in Shangri-La to evaluate the degree of influence of each factor on AGB change. The continuous sample plots from National Forest Inventory (NFI) and Landsat time series were used to estimate the AGB in 1987, 1992, 1997, 2002, 2007, 2012, and 2017. The structural equation model (SEM) was used to analyze the different effects of the four factors on AGB based on five scales: entire, 1987–2002, 2007–2017, low population density, and high population density. The results are as follows: (1) The AGB of *Pinus densata* in Shangri-La decreased from 1987 to 2017, with the total amount falling from 9.52 million tons to 7.41 million tons, and the average AGB falling from 55.49 t/ha to 40.10 t/ha. (2) At different scales, stand structure and climate were the drivers that directly affect the AGB change. In contrast, the agricultural activity had a negative direct effect on the AGB change, and spatial attribute had a relatively small indirect effect on the AGB change. (3) Analyzing the SEM results at different scales, the change of the contribution of the agricultural activity indicates that human activity is the main negative driver of AGB change in Shangri-La, especially at the high population density region. In contrast, the change of the contribution of the stand structure and climate indicates that the loss of old trees has an important influence on the AGB change. Forest resources here and other ecologically fragile areas should be gradually restored by adhering to policies, such as strengthening forest protection, improving forest stand quality, and limiting agricultural production activities.

## 1. Introduction

As the largest ecosystem on earth, the forest ecosystem has various ecological functions such as water conservation, soil and water conservation, and maintenance of biodiversity. Global forest ecosystems can absorb about 861 ± 66 Gt of CO2 per year [1], while aboveground biomass (AGB) accounts for 15% to 30% of the total forest carbon pool [2]. The importance of forest AGB to the whole ecosystem and global carbon cycle [3] means there are many studies on the drivers of forest AGB change. Nevertheless, forest resources are often affected by various factors [4,5], including natural disasters [6,7], climate change [8,9], and human activities [10,11]. Therefore, studying the drivers of forest AGB change in the ecologically fragile area can provide scientific references for the conservation and utilization of forest resources in other regions, especially in the selection of drivers and the establishment of drivers’ relationships.

Forest AGB is influenced by various factors such as spatial attributes [12,13], climate conditions [14,15], stand structure [16,17], and human activities [18,19]. Spatial attributes significantly influence the variation of AGB as they determine the availability of resources for plant growth and survival [20,21]. Climate varies under different elevational and latitudinal gradients [22], with different AGB changes. For example, primary productivity increases with changes in temperature and water use [23]; forest productivity decreases with latitudinal gradient [24]. Moreover, stand structure also has significant effects on the distribution of AGB [17,25]. For example, higher crown density is more conducive to capturing light, which enhances AGB [26]; similarly, different canopy distribution also affects the light and water use efficiency [27,28] and thus, affects AGB. At the same time, agricultural activity also plays an important role in influencing this aspect of the AGB. For example, continuously increased grazing activities lead to an accelerated loss of soil nutrients, which affects vegetation growth [29,30,31]; also, changes in grazing intensity are spatially and temporally correlated with the AGB change [32].

Furthermore, interactions between drivers may result in relationships that already exist at local or regional scales to exhibit different degrees of driving in different environments [19,33]. For example, soil nutrients effectively drive AGB to change at local scales, but at larger spatial scales, the stand structure overshadows the contribution of soil nutrients to AGB [34]. The driving effect of the same driver on the AGB change varies at different scales.

Shangri-La is in the northwestern part of Yunnan Province, at the southeastern end of the Tibetan Plateau. It belongs to the critical water-supporting area in the upper reaches of the Yangtze River, where the environmental bearing capacity in the region is low, and the ecosystem is relatively fragile [35]. Historically, forest resources have been relatively abundant for a long time [36]. However, with the economic development, the forest resources in the Upper Yangtze River region have been decreasing sharply since the 1960s and 1970s, causing more serious soil erosion and directly triggering the 1998 mega-flood in the middle and lower reaches of the Yangtze River. Although the Chinese government strictly enforced the natural forest protection policy at the beginning of the 21st century, it is still tricky for forest resources to recover to the level of the last middle century [37]. Therefore, finding the characteristics and drivers on the AGB change of *Pinus densata*, the major tree species in Shangri-La, from 1987 to 2017, is valuable for forest resource conservation in other ecologically fragile regions.

In this study, we estimate the AGB of *Pinus densata* in different periods based on sample plots from the National Forest Inventory (NFI) (in the year of 1987, 1992, 1997, 2002, 2007, 2012, and 2017) to analyze the spatial and temporal AGB change. The structural equation modeling (SEM) was used to link spatial attribute, climate, stand structure, and agricultural activity to analyze the different degrees of the contribution of each driver on the AGB change according to five scales. Furthermore, the changes in the contribution of each driver at different scales were analyzed.

The objectives of this study were: (1) To analyze the spatial and temporal variation of AGB of *Pinus densata* in Shangri-La. (2) To evaluate the direct and indirect effects of drivers on the AGB change of *Pinus densata* in Shangri-La. (3) To explore the changes in the contribution of each driver at different scales.

## 2. Materials and Methods

### 2.1. Study Area

Shangri-La is part of Yunnan Province in southwest China and in the northwestern part of Yunnan Province, at the southeastern end of the Tibetan Plateau and the eastern side of the Hengduan Mountains (Figure 1b). This area is crisscrossed by ravines and valleys, with an average elevation of 3459 m, decreasing from the northwest to the southeast (Figure 1a). Climate-type here belongs to the mountainous monsoon climate, with an average annual temperature of 5.4 °C, average annual rainfall of 268–945 mm, and a frost-free period of 129–197 days. Due to the sizeable north-south span and noticeable altitude drop, the forest resources in the territory are very rich. Due to its location in the mountainous region, the population in this part is relatively small, and the forest resources have been relatively abundant for a long time, with *Pinus densata*, *Pinus yunnanensis*, and *Abies fabri* being the dominant species [38].

However, due to uncontrolled exploitation of mineral resources, lax control of timber trade, and widespread forest fires, the forest resources here were severely damaged in the last century until 1998 when the state promulgated the natural forest protection in the upper reaches of the Yangtze River. Then, forest ecology was restored. The forest cover increased from 34.4% in 1984 to 76% at present [35,39].

### 2.2. Sample Plots and AGB Estimation

The sample plots used in this study were obtained from the NFI. At the beginning of establishment, each sample plot was distributed according to a 6 × 8 km grid pattern, which was a rectangular sample plot with a size of 28.28 × 28.28 m (each plot was 0.08 ha). The NFI was conducted at 5-year intervals from 1987 to 2017. The multi-year survey produced 136 survey records. Among them, there were 38 permanent sample plots with *Pinus densata* as the dominant tree species.

The variables mainly include land cover types, location (X and Y coordinates), elevation, aspect, slope, soil thickness, average height and percent shrub cover, forest category, origin, dominant tree species, average age, tree diameter at breast height (DBH), tree height, and crown closure. Among the 136 plots, there were 19, 22, 23, 16, 16, 17, and 23 plots for the years 1987, 1992, 1997, 2002, 2007, 2012, and 2017, respectively. Single trees with tree height greater than 1.3 m and diameter at breast height (DBH) greater than 5 cm were selected for each sample plot. 

The amount of AGB in the sample plots was calculated based on average tree height, average DBH, and the number of trees. The sum of the AGB in one sample plot was based on the allometric growth equation of single wood as follows [40,41,42,43]:AGB = 0.073 × DBH^1.739^ × H^0.880^(1)
where: AGB indicates single wood aboveground biomass, DBH is the diameter at breast height, and H is the height of the tree.

### 2.3. Factors Selection and Standardization

The spatial attribute included elevation, relief amplitude, aspect, and latitude (Table 1). The DEM were obtained from the ASTER GDEM dataset provided by the USGS with a spatial resolution of 30 m (https://earthdata.nasa.gov, accessed on 15 August 2021). The latitude was measured by GPS during the field survey. The elevation, relief amplitude, and aspect were extracted from the DEM.

The climate included mean annual precipitation, mean annual temperature, annual sunshine time, relative humidity, and continuous rain days. They were obtained from China Climate Data Network (http://data.cma.cn, accessed on 10 August 2021). These stations of Daocheng, Derong, Xiangcheng, Deqin, Shangri-La, Vixi, and Lijiang were selected. The ordinary kriging method was used to interpolate the climate map within the boundary of Shangri-La.

The stand structure included soil thickness, tree height, mean DBH, stand age and crown density. The data were obtained from the NFI dataset.

The agricultural activity included population density, cultivated land area, forestry investment, and livestock number. Data were obtained from statistical yearbooks published by the local government statistics [36] and the National Bureau of Statistics of China [44].

The dataset was organized by sample number and year. A total of 136 sample plots were collated, each with 18 indices and AGB. All data were standardized [45] using the logarithmic method. The Cronbach’s reliability analysis was performed. Indices with reliability less than 0.6 were excluded, and the remaining indices (Table 2) were used for SEM analysis.

Reliability tests were conducted using the screened indices. The Cronbach’s alpha coefficient was obtained as 0.687. It can prove that the data quality was acceptable [46,47]. 

### 2.4. Establishment and Analysis of SEM

We assumed that spatial attribute directly drove the changes of climate and agricultural activity through latitude, elevation, and topography [18,21]. While different climate conditions influenced the stand structure, thus, directly and indirectly, determining the AGB change [14,26,48]. While the stand structure and agricultural activity directly influenced the AGB change [17,20,49] (Figure 2). 

Based on this assumption, the SEM model was established using the four factors, ten indices, and the AGB. In this study, four statistical parameters: comparative fit index (CFI), goodness of fit index (GFI), standardized root mean square residual (SRMR), and chi-squared degrees of freedom ratio (Chi-square/df), were used to evaluate the fitting accuracy of the model.

### 2.5. Scales of SEM Analysis

In order to investigate the drivers, five scales were used in this study:(1)The entire scale: All sample plots were used for SEM analysis.(2)The 1987–2002 scale: The sample plots from 1987, 1992, 1997, and 2002 were classified as one scale. The number of sample plots was 69.(3)The 2007–2017 scale: The sample plots from 2007, 2012, and 2017 were classified as another scale. The number of sample plots was 67.(4)The low population density scale: The townships with population density ≤ 15 people/km^2^ were classified as the low population density townships (Table 3).(5)The high population density scale: The townships with population density > 15 people/km^2^ were classified as the high population density townships.

## 3. Results

### 3.1. Spatial and Temporal Variation of AGB of Pinus densata in Shangri-La

The estimated results of AGB of *Pinus densata* in Shangri-La are shown in Table 4.

Between 1987 and 2002, the distribution area was a decreasing trend, while during 2002 and 2017, the distribution area increased from 170,589.86 ha to 184,815.84 ha. The average AGB continued to decrease from 55.49 t/ha in 2017 to 40.10 t/ha in 2017.

The total AGB showed a trend of 2.11 million tons, with an average decrease of 0.07 million tons per year. The most significant decrease in AGB occurred between 1992 and 1997, with a reduction of 1.14 million tons. The smallest decrease in AGB occurred between 2012 and 2017, with a decrease of 0.05 million tons.

Most of the *Pinus densata* in Shangri-La are distributed in Jiantang, Geza, Dongwang, and Nixi Township (Figure 3 and Figure 4). These four townships accounted for about 70% of the total distribution area, but the proportion of their total AGB decreased from 76% in 1987 to 67% in 2017. Except for Sanba and Tiger Leaping Gorge Township, where the AGB was increased between 2012 and 2017, the total AGB of all the townships showed a trend of yearly decrease. Among them, the total AGB of Geza, Dongwang, Jiantang, and Nixi Township reduced larger than other townships.

### 3.2. SEM Results

The SEM model at five scales explains 65.2%, 43.3%, 88.3%, 75.8%, and 88.1% of the AGB variation, respectively. The four indicators (CFI, GFI, SRMR, and Chi-square/df) were within reasonable ranges. The R^2^, path coefficients, and significance show that the SEM model performs well for the analysis at all scales (Figure 5). 

At the entire scale, spatial attribute directly affected climate and agricultural activity (0.624 and 0.802, for short, the numbers after the factors represent path coefficients). The climate, stand structure, and agricultural activity all directly affected AGB (0.378, 0.760, and −0.227). In comparison, spatial attribute had a significant indirect effect on stand structure through climate (−0.023) and had a significant indirect effect on AGB through stand structure (−0.017). 

At the 1987–2002 scale, spatial attribute directly affected climate and agricultural activity (0.680 and 0.659). The climate, stand structure, and agricultural activity directly affected AGB (0.390, 0.986, and −0.270). In comparison, spatial attribute had a significant indirect effect on stand structure through climate (path coefficient of −0.119) and had a significant indirect effect on AGB through stand structure (−0.117). Furthermore, spatial attribute had a significant indirect effect on AGB through agricultural activity (−0.177).

At the 2007–2017 scale, spatial attribute directly affected climate and agricultural activity (0.672 and 0.743). The climate, stand structure, and agricultural activity directly affected AGB (0.408, 0.545, and −0.226). In contrast, spatial attribute had a significant indirect effect on stand structure through climate (−0.101) and had a significant indirect effect on the AGB through stand structure (−0.055). At the same time, spatial attribute also had a significant indirect effect on AGB through agricultural activity (−0.168).

At the low population density scale, spatial attribute directly affected climate and agricultural activity (−0.820 and 0.440). The climate and stand structure directly affected AGB (0.361 and 0.686). In contrast, spatial attribute had a significant indirect effect on stand structure through climate (−0.257) and had a significant indirect effect on AGB through stand structure (−0.177). However, in this model, since the direct effect of agricultural activity on AGB was not significant (−0.102), the indirect effect of spatial attribute on AGB through agricultural activity was also not significant (−0.045).

At the high population density scale, spatial attribute directly affected climate and agricultural activity (−0.843 and 0.857). The climate, stand structure, and agricultural activity directly affected AGB (0.716, 0.636, and −0.933). In contrast, spatial attribute had a significant indirect effect on stand structure through climate (−0.212) and had a significant indirect effect on AGB through stand structure (−0.135). Furthermore, spatial attribute had a significant indirect effect on AGB through agricultural activity (−0.799).

Overall, the effect of stand structure and climate is significantly larger than the effect of other factors on the AGB change. Although the effect of agricultural activity is small, it is the only factor with a negative direct impact. The spatial attribute has a significant indirect effect on the AGB change through the direct effect of climate and agricultural activity.

### 3.3. Contribution Changes of Drivers at Different Scales 

In the SEM model, the total effect is the sum of the direct and indirect effects. The relative contribution and path coefficient of each factor under different scales are shown in Figure 6. 

At the entire scale, the factors were ranked according to the total effect on the AGB change: stand structure, climate, agricultural activity, and spatial attribute (0.760, 0.349, −0.227, and 0.036, respectively). At the 1987–2002 scale, the factors were ranked: stand structure, climate, agricultural activity, and spatial attribute (0.986, 0.270, 0.217, and 0.030, respectively). At the 2007–2017 scale, the factors were ranked: stand structure, climate, agricultural activity, and spatial attribute (0.545, 0.489, 0.226, and 0.161, respectively). At the low population density scale, the factors were ranked: stand structure, climate, spatial attribute, and agricultural activity (0.686, 0.576, 0.518, and 0.102, respectively). At the high population density scale, the factors were ranked: agricultural activity, climate, stand structure, and spatial attribute (0.933, 0.876, 0.636, and 0.061, respectively).

The contribution of each driver changed significantly from the 1987–2002 scale to the 2007–2017 scale. The contribution of stand structure decreased from 66% to 38%; the contribution of climate increased from 14% to 35%; the contribution of spatial attribute increased from 2% to 11%. The percentage change of agricultural activity was minor.

The contribution of each driver also changed significantly under the low population density scale and high population density scale. Under the low population density scale, stand structure, climate, and spatial attribute were the main drivers (contributions of 36%, 31%, and 28%, respectively), while the contribution of agricultural activity was less. However, at the high population density scale, agricultural activity, climate, and stand structure were the main drivers (contributions of 37%, 35%, and 25%, respectively), while the contribution of spatial attribute decreased from 28% to 3%.

## 4. Discussion

### 4.1. Comparison of AGB Estimation

One goal of this study was to estimate the AGB of *Pinus densata* from 1987 to 2017 and analyze its temporal and spatial variation. In order to compare and validate the results of this study, we summarized the results of biomass estimation in this region by various researchers [41,42,50,51,52,53] (Table 5).

Usually, the aboveground biomass accounts for about 65–88% of the tree layer biomass; the aboveground biomass accounts for about 15–30% of the forest biomass, and the underground biomass accounts for about 4–8% [54,55]. It is deduced that the AGB in 2008 from Yue’s study is between 9.21 million and 12.48 million tons; the AGB in 2009 from Cheng’s study is between 4.93 million and 6.67 million tons; the AGB in 2009 from Wang’s study is between 3 million and 6 million tons. The AGB in 2014 from Sun’s study is 11.72 million tons; the AGB in 2015 in 2014 from Xie’s study is 8.9 million tons; the AGB in 1987,1997 and 2007 from Zhang’s study is 7.78 million tons,8.42 million tons, and 7.65 million tons, respectively. Compared with the above results, the AGB of *Pinus densata* derived from this study is in a reasonable range.

Shangri-La is located at the southeastern end of the Tibetan Plateau. Due to the natural environment, religious beliefs, the local population prefers wooden houses [56,57,58]. It takes around 120 mature trees to build a traditional timber-framed Tibetan house [56,59]. The huge local demand for timber has led to frequent unlawfully logging. Although the Chinese government has made great efforts to protect forest resources [35,38]. However, the loss of the old trees here will cause a continued reduction in AGB. [41,42,50]. Moreover, the construction of expressways, trains, and high-speed railways since the 21st century has also occupied much forest land. These reasons led to a decrease in AGB.

### 4.2. Analysis of Drivers on the AGB Change

Compared with the SEM results under different scales, it can be found that stand structure and climate were the main driving factors affecting the AGB change. In contrast, the agricultural activity and spatial attribute were less influential. Compared with other scholars’ results of related studies, the main driver of AGB changes in this study was reasonable [20,26,60,61].

Stand Structural has been shown that it could significantly influence the AGB in some studies [48,49,62]. Xiang et al. reported that stem density significantly affected the biomass of a subtropical mountain moist forest [63]. Liu et al. reported that the DBH might affect the AGB [17]. Nogueira et al. also reported that stand age has strong effects on AGB [64]. The continued decrease in AGB is strongly correlated with the continued growth of young stands in the forest [65]. 

Climate mainly explains the ability of vegetation to obtain light and water [66,67]. Previous studies have shown that climate directly affects stand structure. The temperature and moisture indirectly affect AGB through single plant size [19]. Amit et al. also reported that solar radiation and temperature might impact the growth of vegetation.

Agricultural activity also showed a strong negative effect on the AGB change in the long term [30,32]. The expansion of farmland is accompanied by a continuous decrease in forest and grassland, which directly leads to a decrease in AGB. In addition, stockbreeding livestock farming is prevalent in the upper Yangtze River region, and stockbreeding livestock activities have a strong negative impact on forest resources [68,69], which leads to a decrease in AGB [70,71].

Spatial attribute has important effects on plant growth, community structure, and ecosystem processes [61,72]. Related studies have shown that increasing elevation leads to a decrease in humidity and temperature and an increase in precipitation or snowfall, which in turn have an impact on the AGB change [62,73]; the greater the topographic relief, the lower the AGB; and the AGB decreases significantly with increasing elevation.

In the current study, the selection of drivers is still somewhat problematic [19,25]. Some scholars believe that species diversity has a more significant influence on AGB [17]. A variety of indices were selected [18,26,61,74] for analysis (e.g., richness index, Shannon diversity, Simpson diversity, etc.). However, the main tree species in Shangri-La are primarily pure forests [38]. The dominant tree species of NFI plots used in the study are *Pinus densata*. In addition, forest pest and disease hazards and forest fires have a greater impact on forest resources [4,19,35], but these factors were not used in this study due to the lack of relevant data. Some scholars also believe that soil properties have a more significant influence on AGB [5,21] for analysis (e.g., pH, organic matter, total nitrogen, total phosphorus, etc.), but our research is on a long term scale that the soil property did not change too much. In the future, more factors should be explored to enhance the analysis.

### 4.3. Changes in the Contribution of Each Driver at Different Scales

In previous studies, most scholars have mentioned that the extent of drivers varies at different scales [4,19,60]. Most scholars have focused their work on finding the main drivers [5,61]. In this research, long time series data were used to analyze at multiple scales to identify changes in drivers.

By comparing the changes in factor contribution from the 1987–2002 scale to the 2007–2017 scale, it can be found that the contribution of stand structure and agricultural activity decreased, while the contribution of climate and spatial attribute increased.

From 1987 to 2002, old trees were heavily logged. The top 5% of large trees in the stand, however, contributed more than 60% of the stand’s AGB, and the loss of old trees was a direct cause of the decline in stand structure [19,61,74]. The 2% reduction in the contribution of agricultural activity is mainly reflected in the return of farmland located on steep slopes to forests and grasslands. Climate has less influence on 20 or 30 years-old trees but has a greater influence on the growth process of new trees, so its contribution increases from 14% to 35%. The increase in spatial attributes is mainly due to their indirect influence on AGB change through climate.

By comparing the changes in factor contribution from the low population density scale to the high population density scale, it can be found that the contribution of agricultural activity increased; while the spatial attribute and stand structure decreased, the climate did not change much.

The intensity of agricultural activities in low population density townships is smaller than that in high population density townships, resulting in a larger difference in the contribution of agricultural activity. The low population density towns are generally located in areas of high altitude and relatively large undulations, so the moisture conditions are relatively poor, while the opposite is true for the high population density towns. Therefore, the contribution of spatial attribute is reduced. Forests in low population density townships are less affected by humans, so the contribution of stand structure is higher. 

Although five scales have been used in this study for the analysis. Due to the lack of sample plots, this study could not perform further differentiation analysis while ensuring high accuracy of SEM results. There are still more scales of analysis that can be used in future studies, such as different spatial attributes (elevation, aspect), climatic conditions (precipitation, temperature, humidity), and the origin of forests (natural forest, secondary forest).

## 5. Conclusions

From 1987 to 2017, the total AGB of *Pinus densata* in Shangri-La decreased from 9.52 million tons to 7.41 million tons; the average AGB decreased from 55.49 t/ha to 40.10 t/ha. The SEM results showed that stand structure and climate were the significant drivers, and then agricultural activity and spatial attribute. Loss of old trees, climatic effects on new trees, and grazing activities were the main driver on the AGB change. In addition, SEM models can obviously distinguish the changes in the degree of influence by different factors when studying at a subdivided scale. This study could provide a reference for other scholars to select drivers and establish drivers’ relationships. Forests managers could also take appropriate decisions to protect forest resources in other similar ecological regions.

## Figures and Tables

**Figure 1 ijerph-19-00400-f001:**
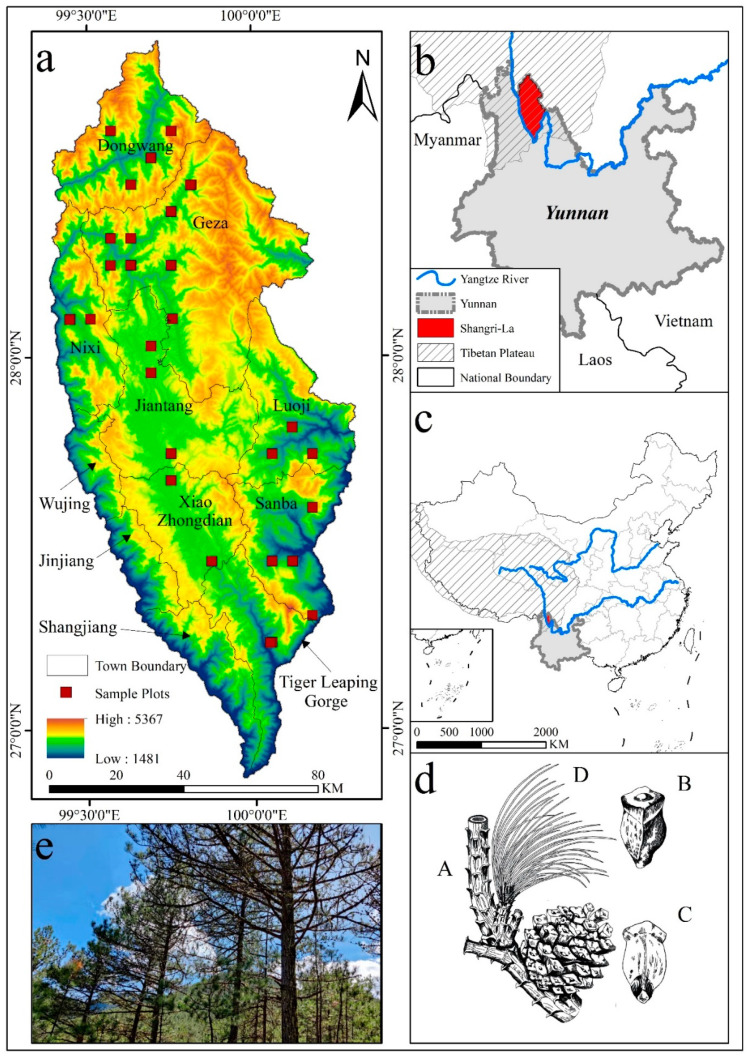
Study area and the distribution of sample plots: (**a**) Distribution of sample plots (**b**) Location of Shangri-La (**c**) Location of Yunnan in China (**d**) Morphological characteristics of *Pinus densata*, where A represents the Cone-bearing branchlet and seed cone, B represents the seed scale abaxial view, C represents the seed scale adaxial view, and D represents conifer pine needles. (**e**) Photo of *Pinus densata* in one sample plot.

**Figure 2 ijerph-19-00400-f002:**
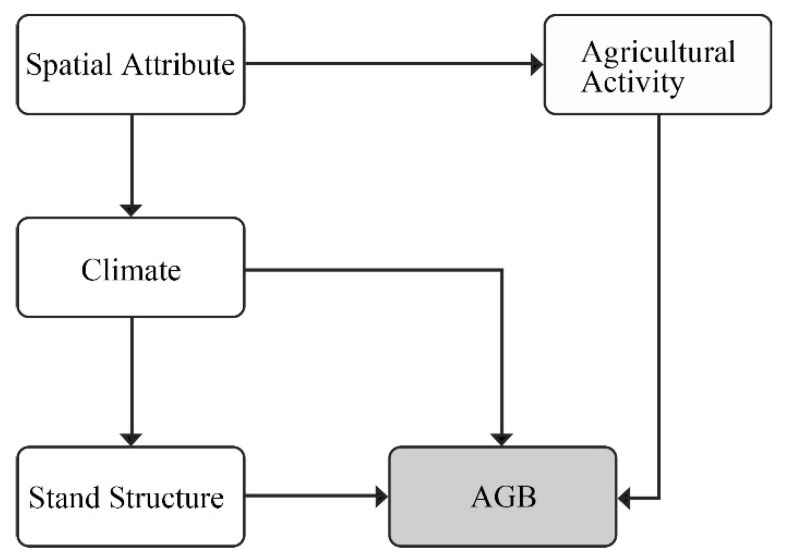
Conceptual model of SEM.

**Figure 3 ijerph-19-00400-f003:**
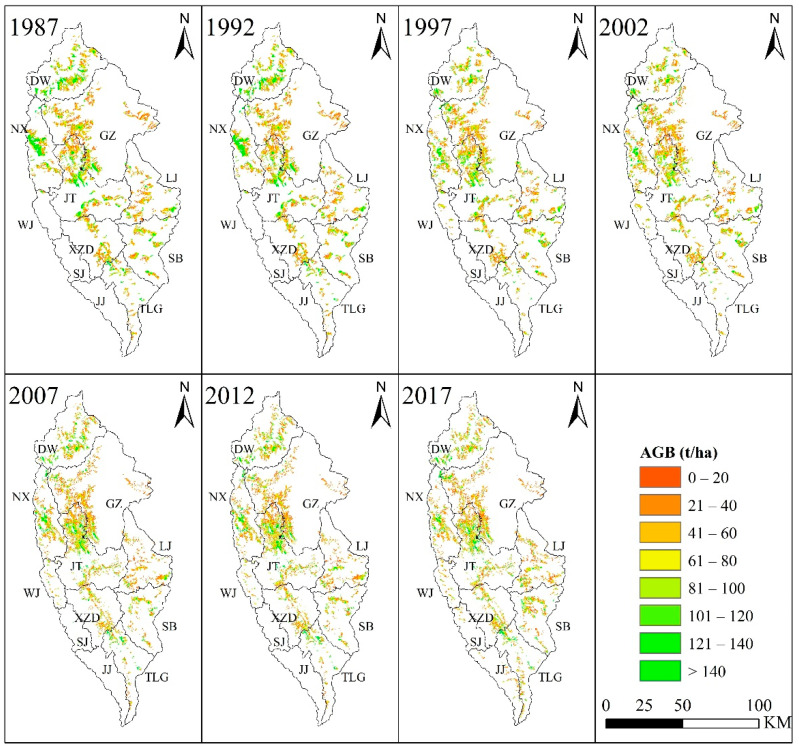
AGB estimation results during the period of 1987 to 2017. DW, GZ, TLG, JT, LJ, NX, SB, XZD, WJ, JJ, SJ are short for Dongwang, Geza, Tiger Leaping Gorge, Jiantang, Luoji, Nixi, Sanba, Xiao Zhongdian, Wujing, Jinjiang, Shangjiang, respectively.

**Figure 4 ijerph-19-00400-f004:**
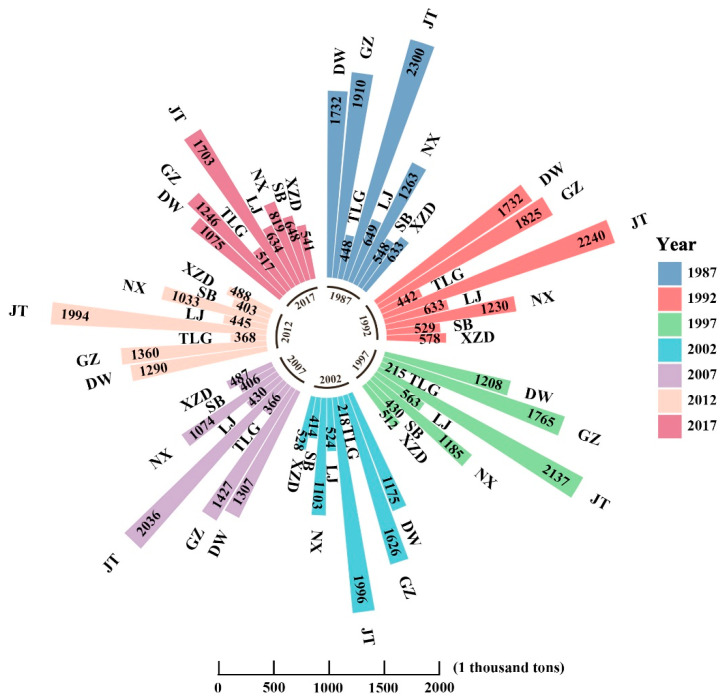
AGB of each town from 1987 to 2017. DW, GZ, TLG, JT, LJ, NX, SB, XZD are short for Dongwang, Geza, Tiger Leaping Gorge, Jiantang, Luoji, Nixi, Sanba, Xiao Zhongdian, respectively.

**Figure 5 ijerph-19-00400-f005:**
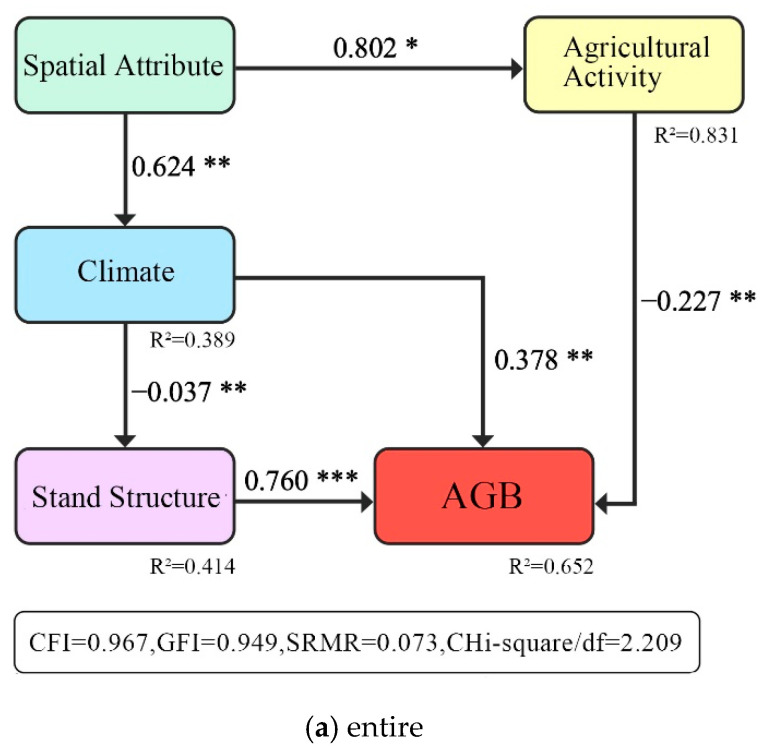
The SEM results can reveal four drivers’ direct and indirect effects on AGB changes of *Pinus densata*. Arrows indicate pathways, numbers next to arrows are normalized path coefficients, and * indicates significance levels (significance levels are as follows: ***, *p* < 0.001; **, *p* < 0.01; *, *p* < 0.1). R^2^ represents the coefficient of determination of the endogenous latent variable. (**a**–**e**) describe the results of SEM models at different scales, respectively.

**Figure 6 ijerph-19-00400-f006:**
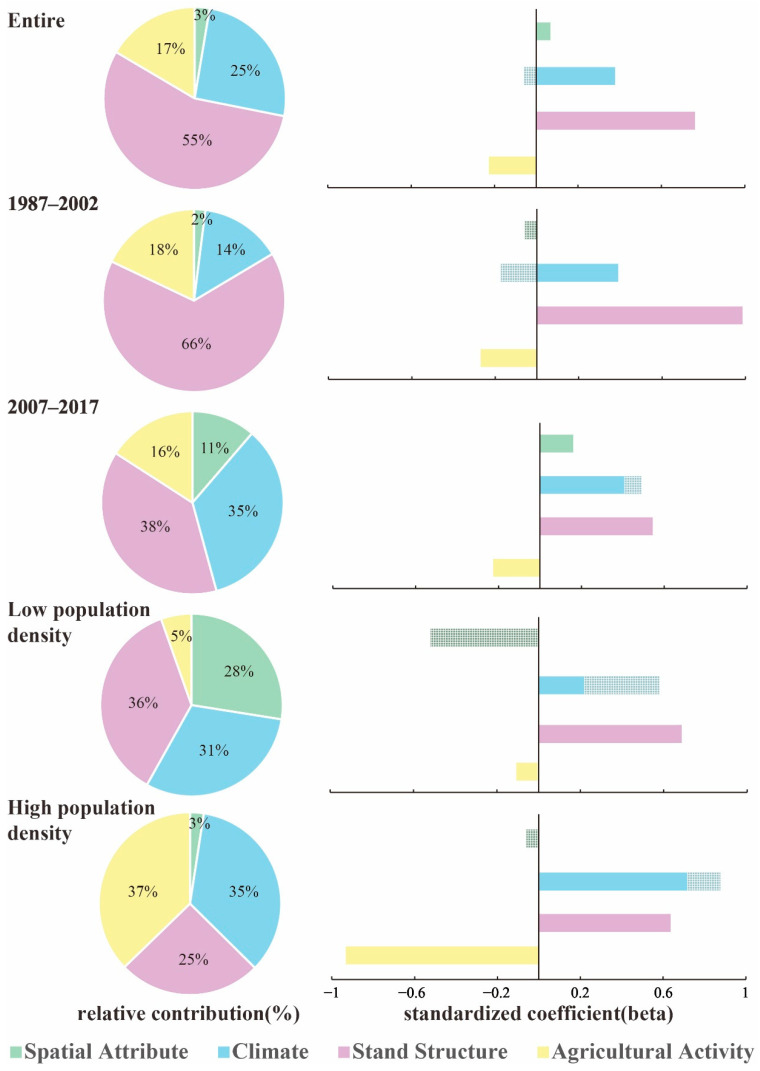
Standardized coefficient and relative contribution of the four drivers on the AGB changes. The pies show the relative importance of each driver on the AGB changes. The filled bars indicate the direct effect, and the striped bars indicate the indirect effect.

**Table 1 ijerph-19-00400-t001:** Descriptions of variables.

Variables	Unit	Mean Value	Coefficient of Variation
Spatial Attribute			
Elevation	m	3286.78	7.85%
Relief Amplitude	°	39.88	53.85%
Aspect	°	155.30	62.01%
Latitude	°	27.98	1.54%
Climate			
Mean Annual Precipitation	mm	729.12	17.24%
Mean Annual Temperature	°C	9.14	16.44%
Annual Sunshine Time	h	2081.41	4.74%
Relative Humidity	%	62.87	6.37%
Continuous Rain Days	day	130.05	9.24%
Stand Structure			
Soil Thickness	cm	58.24	38.46%
Tree Height	m	17.83	78.07%
Mean DBH	cm	10.22	47.40%
Stand Age	year	53.69	66.86%
Crown Density	%	56.83	31.21%
Agricultural Activity			
Population Density	person/km^2^	11.08	68.86%
Cultivated Land Area	ha	17,923.66	63.36%
Forestry Investment	yuan	1537.24	79.89%
Livestock Number	number	22,115.67	77.75%

**Table 2 ijerph-19-00400-t002:** List of driving factors.

Factor Type	Indices
Spatial Attribute	Elevation
	Relief Amplitude
Climate	Annual Sunshine Time
	Relative Humidity
	Continuous Rain Days
Stand Structure	Tree Height
	Mean DBH
	Stand Age
Agricultural Activity	Cultivated Land Area
	Livestock Number

**Table 3 ijerph-19-00400-t003:** The number of high/low population density towns and corresponding sample plots.

High Population Density Towns		Low Population Density Towns	
Town	Number of Sample Plots	Town	Number of Sample Plots
Tiger Leaping Gorge	7	Nixi	6
Xiao Zhongdian	13	Luoji	16
Jiantang	15	Dongwang	22
Sanba	19	Geza	38
Total	54		82

**Table 4 ijerph-19-00400-t004:** Results of estimation of AGB of *Pinus densata* by years.

Year	Area of *Pinus densata* (ha)	Average AGB (t/ha)	Total AGB (Million Tons)
1987	171,560.28	55.49	9.52
1992	171,560.28	53.89	9.25
1997	170,589.86	47.52	8.11
2002	170,589.86	45.02	7.68
2007	174,179.37	43.76	7.62
2012	174,213.12	42.84	7.46
2017	184,815.84	40.10	7.41

**Table 5 ijerph-19-00400-t005:** Summary of results by different researchers.

Researcher	Subject	Study Year	Estimated Biomass (Million Tons)	AGB (Million Tons)
Cairong Yue [50]	Tree layer biomass	2008	14.18	9.21–12.48
Pengfei Cheng [51]	Tree layer biomass	2009	7.58	4.93–6.67
Jinliang Wang [52]	Forest biomass	2009	20.00	3–6
Xuelian Sun [41]	AGB	2014		11.72
Fuming Xie [53]	AGB	2015		8.9
		1987		7.78
Jialong Zhang [42]	AGB	1997		8.42
		2007		7.65
		1987		9.52
		1992		9.25
		1997		8.11
This study	AGB	2002		7.68
		2007		7.62
		2012		7.46
		2017		7.41

Note: Forest biomass include the tree layer biomass, shrub layer biomass, herb layer biomass, litter layer biomass, and soil layer biomass; Tree layer biomass includes the aboveground biomass and the underground biomass.

## Data Availability

The data presented in this study are available on request from the corresponding author. The data are not publicly available due to the NFI dataset is confidential.

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
