# Peer review of "Temporal and Spatial Variation of Aboveground Biomass of Pinus densata and Its Drivers in Shangri-La, CHINA"

_ijerph, 2021, doi:10.3390/ijerph19010400_

Round 1

Reviewer 1 Report

The paper is presenting an interesting approach to the understanding of the variation of AGB.

Many data, elaborations and information are presented.

However the organization of the text and the presentation of the results is not always easy to understand and would require a rationalization and a more thorough use of explanatory tables and graphs.

Some specific indication are here listed:

WHERE (W)                 

ERRATA (E)                

CORRIGE (C)

W - Abstract - rows 14-17       

E -  the “-“ breaks some words                

C - Erase non necessary “-“

W - Abstract - row 23               

E - Pinus densata                  

C - Please in Italic format 

W - FIGURE 1 pag. 4               

E - d) B represents the abaxial seed             

C - d) B represents the seed scale abaxial view

W - FIGURE 1 pag. 4               

E - d) C represents the adaxial seed             

C - d) C represents the seed scale adaxial view

W - TABLE 1 pag. 5                 

E - some parameters obtained from the NFI dataset have to be better explained as they are so important for the research results

W - TABLE 1 pag. 5 

E - measure units in relief amplitude an aspect       

C - no understanding the absence of measure units

In my opinion the theory and the methodology of the paper are interesting, although probably difficult to be used in geographical and environmental contexts other than the Chinese ones described in the paper. This is the reason because I thought that the paper is not extremely interesting.
Methodology seems good. As I wrote some parameters could be better presented.

Concerning the presentation of the results nothing is really wrong but paper can be improved by lightening  the form.
E.g.: the table n.4 can be easily and efficiently transformed in a chart, the rows 181 - 183 are a trivial rereading of table no. 4, many of the numbers in paragraphs 3.2 and 3.3 can be used to build some much more understandable and immediate charts, the "drivers on AGB change" in paragraph 4.2 (rows 314-318) could be converted in a chart or in a table to be more readable..... and so on.

Author Response

Reviewer 1:

Dear Reviewer,

Thanks very much for taking the time to review this manuscript. I really appreciate all your comments and suggestions! Please find my itemized responses below and my revisions/corrections in the re-submitted files.

In the following, we transcribe the comments in gray and our response in red in a response section.

W - Abstract - rows 14-17

E - the “-” breaks some words

C - Erase non necessary “-”

Response: I am very sorry for this writing error, which has been corrected. The writing format of this manuscript has been checked.

W - Abstract - row 23

E - Pinus densata

C - Please in Italic format

Response: This error has been corrected. The "Pinus densata" format in the full text has been checked.

W - FIGURE 1 pag. 4              

E - d) B represents the abaxial seed            

C - d) B represents the seed scale abaxial view

W - FIGURE 1 pag. 4              

E - d) C represents the adaxial seed            

C - d) C represents the seed scale adaxial view

Response: I am very sorry for any confusion I have caused due to my poor knowledge of botany. I have made the changes as you suggested.

W - TABLE 1 pag. 5                

E - some parameters obtained from the NFI dataset have to be better explained as they are so important for the research results

Response: I agree with you and have already made additional introductions to the relevant parameters.

W - TABLE 1 pag. 5

E - measure units in relief amplitude an aspect      

C - no understanding the absence of measure units

Response: I am very sorry for this mistake due to my unclear explanation.

R=Hmax-Hmin   (1)

The formula for calculating relief amplitude is shown in (1), which  is the relief amplitude,  is the maximum value in the region,  is the minimum value in the region. This indicator is a ratio, so I did not initially add a measure unit.

However, the absence of the measure unit is confusing, so that I will use ° as the measure unit of relief amplitude and aspect.

In my opinion the theory and the methodology of the paper are interesting, although probably difficult to be used in geographical and environmental contexts other than the Chinese ones described in the paper. This is the reason because I thought that the paper is not extremely interesting.

Response: I agree with you that the previous statement was indeed too extreme.

Most of the research cases in the literature that I have read were Chinese at the beginning of the experimental design [1,2,3]. So, this led me to believe that the approach was also applicable to other regions.

However, I believe that similar studies can be carried out even in other regions if the right factors are selected, and the interactions between the factors are clearly understood [4,5,6].

Anyway, I agree with you and have made changes to the relevant parts of the manuscript.

[1] Zhang, H.; Wang, K.; Zeng, Z.; Zou, Z.; Xu, Y.; Zeng, F. Multiple Factors Drive Variation of Forest Root Biomass in Southwestern China. Forests 2018, 9, 456. https://doi.org/10.3390/f9080456

[2] Yuan, Z., Ali, A., Jucker, T., Ruiz-Benito, P., Wang, S., Jiang, L., … Loreau, M. (2019). Multiple abiotic and biotic pathways shape biomass demographic processes in temperate forests. Ecology, e02650. doi:10.1002/ecy.2650

[3] Ali, A., Chen, H. Y. H., You, W.-H., & Yan, E.-R. (2019). Multiple abiotic and biotic drivers of aboveground biomass shift with forest stratum. Forest Ecology and Management, 436, 1–10. doi: 10.1016/j.foreco.2019.01.007

[4] Gutiérrez-Girón, A., Rubio, A., & Gavilán, R. G. (2013). Temporal variation in microbial and plant biomass during summer in a Mediterranean high-mountain dry grassland. Plant and Soil, 374(1-2), 803–813. doi:10.1007/s11104-013-1887-6

[5] Chun, J.-H., Ali, A., & Lee, C.-B. (2020). Topography and forest diversity facets regulate overstory and understory aboveground biomass in a temperate forest of South Korea. Science of The Total Environment, 140783. doi: 10.1016/j.scitotenv.2020.1407

[6] Holdaway, R. J., Easdale, T. A., Carswell, F. E., Richardson, S. J., Peltzer, D. A., Mason, N. W. H., … Coomes, D. A. (2016). Nationally Representative Plot Network Reveals Contrasting Drivers of Net Biomass Change in Secondary and Old-Growth Forests. Ecosystems, 20(5), 944–959. doi:10.1007/s10021-016-0084-x

Methodology seems good. As I wrote some parameters could be better presented.

Response: I agree with your comments and have already made additional explanations of the parameters.

Concerning the presentation of the results nothing is really wrong but paper can be improved by lightening the form.

E.g.: the table n.4 can be easily and efficiently transformed in a chart, the rows 181 - 183 are a trivial rereading of table no. 4.

Response: I agree with you and have made changes to the relevant parts of the manuscript.

Many of the numbers in paragraphs 3.2 and 3.3 can be used to build some much more understandable and immediate charts.

Response: Since this study uses five different scales for the analysis, this makes the results enriched. And Figure5 and Figure6 correspond to the numbers in paragraphs 3.2 and 3.3 respectively. If we did not use these numbers for interpretation, the results would be unclear.

Other scholars [7,8,9] who have used SEM models for their analyses have also written in this general pattern.

[7] Liu, L.; Zeng, F.; Song, T.; Wang, K.; Du, H. Stand Structure and Abiotic Factors Modulate Karst Forest Biomass in Southwest China. Forests 2020, 11, 443. https://doi.org/10.3390/f11040443

[8] Li, N., Wang, J., Yin, W., Jia, H., Xu, J., Hao, R., … Shi, Z. (2021). Linking water environmental factors and the local watershed landscape to the chlorophyll a concentration in reservoir bays. Science of The Total Environment, 758, 143617. doi: 10.1016/j.scitotenv.2020.143617

[9] Ali, A., Lin, S., He, J., Kong, F., Yu, J., & Jiang, H. (2019). Elucidating space, climate, edaphic and biodiversity effects on aboveground biomass in tropical forests. Land Degradation & Development. doi:10.1002/ldr.3278

The "drivers on AGB change" in paragraph 4.2 (rows 314-318) could be converted in a chart or in a table to be more readable.... and so on.

Response: I agree with you. The content of lines 314-318 was a repetitive description of the results, and this paragraph has been removed.

Thanks again for your careful review and positive advice!

Reviewer 2 Report

The text is well written and presents relevant information that can be used as a basis for conservation studies of the area as well as studies of other organisms such as fungi and animals that inhabit the aboveground biomass of forests. Below are some suggestions for improving the text.

Title: Temporal and spatial variation of aboveground biomass of Pinus densata Masters and its drivers in Shangri-La, China

Line 15: ...incredibly fragile...

Line 16: ...Pinus densata...(in italics)

Lines 17-18: ...spatial attribute, climate, stand structure, and agricultural activity...

Line 23: ...Pinus densata...

Line 25: ...At different scales, stand structure and climate...

Lines 26-27: ...agricultural activity had a negative direct effect on the AGB change, and spatial atribute..

Lines 28-29: ...agricultural activity...

Line 31: ...stand structure and climate...

Keywords: Do not use names included in title as keywords.

Lines 84-85:...stand structure and climate...

Line 97: (...with an average elevation of 3459 m and an altitude difference of 4042 m...) this sentence is not clear. I suggest to include directly the altitude variation.

Figure 1. The figure is well-crafted and beautiful. But, looking at the figure of letter b it is not possible to visualize Yunnan in China or world. I suggest including a map where you can view Yunnan in China.

Lina 116: ...28.28×28.28 m...

From material and methods, the words Agricultural Activity, Spatial Attribute, Stand Structure and Climate are written with the first letter capitalized. I suggest using the first letter lowercase as in the introduction.

Author Response

Reviewer 2:

Dear Reviewer,

Thanks very much for taking the time to review this manuscript so carefully. I really appreciate all your comments and suggestions! Please find my itemized responses below and my revisions/corrections in the re-submitted files.

In the following, we transcribe the comments in gray and our response in red in a response section.

Line 15: incredibly fragile...

Response: I am very sorry for this writing error, which has been corrected. The writing format of this manuscript has been checked.

Lines 16: Pinus densata... (in italics) …

Line 23: Pinus densata...

Response: These two errors has been corrected. The "Pinus densata" format in the full text has been checked.

Lines 17-18: spatial attribute, climate, stand structure, and agricultural activity...

Line 25: At different scales, stand structure and climate...

Lines 26-27: ...agricultural activity had a negative direct effect on the AGB change, and spatial attribute...

Lines 28-29: ...agricultural activity...

Line 31: ...stand structure and climate...

Lines 84-85: stand structure and climate...

From material and methods, the words Agricultural Activity, Spatial Attribute, Stand Structure and Climate are written with the first letter capitalized. I suggest using the first letter lowercase as in the introduction.

Response: Thank you very much for your careful reading of my manuscript. I agree with your suggestions. I have corrected all "Spatial Attribute", "Climate", "Stand Structure" and "Agricultural Activity" to lowercase.

Keywords: Do not use names included in title as keywords.

Response: I agree with your suggestion. The original keywords have been amended.

Line 97: (...with an average elevation of 3459 m and an altitude difference of 4042 m...) this sentence is not clear. I suggest to include directly the altitude variation.

Response: I agree with your suggestion and have corrected the relevant parts of the manuscript.

Figure 1. The figure is well-crafted and beautiful. But, looking at the figure of letter b it is not possible to visualize Yunnan in China or world. I suggest including a map where you can view Yunnan in China.

Response: Your suggestion is very beneficial to me because most people are not aware of the exact location of Yunnan Province in China. A map of the location of Yunnan in China has been added.

Lina 116: ...28.28×28.28 m...

Response: This error has been corrected.

Thanks again for your careful review and positive advice!

This manuscript is a resubmission of an earlier submission. The following is a list of the peer review reports and author responses from that submission.